Characteristics of Central American brocket deer resting sites in a tropical mountain cloud forest in eastern Mexico

Muñoz Vazquez Brenda 1 2
Gallina Tessaro Sonia 3
León-Paniagua Livia llp@ciencias.unam.mx 1
1 Museo de Zoología “Alfonso L. Herrera”, Departamento de Biología Evolutiva, Facultad de Ciencias, Universidad Nacional Autónoma de México , Mexico City , Mexico City , Mexico
2 Posgrado en Ciencias Biológicas edificio D, 1ºPiso. Circuito de Posgrados, Ciudad Universitaria Del. Coyoacan. C. P. 04510, Universidad Nacional Autónoma de México , Ciudad de México , México
3 Red de Biología y Conservación de Vertebrados, Instituto de Ecología, A.C , Xalapa , Mexico
Hedrick Ann
Electronic publication date: 2022 Jan 3
Publication date: 2022
Volume: 10
Electronic Location ID: e12587
Received 2021 May 18; Accepted 2021 Nov 11
Copyright: ©2022 Muñoz Vazquez et al.
Copyright year: 2022
Copyright holder: Muñoz Vazquez et al.
License: This is an open access article distributed under the terms of the Creative Commons Attribution License, which permits unrestricted use, distribution, reproduction and adaptation in any medium and for any purpose provided that it is properly attributed. For attribution, the original author(s), title, publication source (PeerJ) and either DOI or URL of the article must be cited.
License URL: https://creativecommons.org/licenses/by/4.0/

Keywords: Con cursivas, Habitat specialist, Central American brocket deer, Resting sites, Mountain cloud forest, Resting site, Microhabitat, Landscape

Funding: UNAM-PAPIIT IN222019 and Rufford Foundation CONACyT This work was supported by UNAM-PAPIIT IN222019 and Rufford Foundation Small Grants. Brenda Muñoz Vazquez received a scholarship from CONACyT. The funders had no role in study design, data collection and analysis, decision to publish, or preparation of the manuscript.

==============================
The Central American brocket deer is a vulnerable species. Geographically isolated populations have been affected by poaching and habitat fragmentation, leading to local extinctions. It is therefore important to understand this species’ habitat characteristics, particularly of resting sites, which play a crucial role in survival and fitness. We describe the characteristics and distribution patterns of Central American brocket deer resting sites at the microhabitat and landscape scales in San Bartolo Tutotepec, Hidalgo, México. We conducted eight bimonthly field surveys between November 2017 and March 2019, consisting of 32 transects of 500 m length to search for fecal pellets, footprints, scrapes, and browsed plants. At each resting site we identified, we measured canopy closure, horizontal thermal cover, protection from predators for fawns and adults, escape routes, slope from the ground, presence of scrapes, cumulative importance value of the edible plant species, and distance from the resting site to the nearest water resource to characterize the site at the microhabitat scale. At the landscape scale, we identified the type of biotope, elevation, aspect, and slope. We compared all of these parameters from resting sites with a paired randomly selected site to serve as a control. We performed a multiple logistic regression to identify the parameters associated with the resting sites and a point pattern analysis to describe their distribution. We characterized 43 resting sites and their corresponding control plots. At the microhabitat scale, resting sites were associated with higher vertical thermal cover, more concealment cover, more escape routes, more edible plant species, higher slope from the ground, and closer distance to water resources. At the landscape scale, resting sites were associated with beech forest, oak forest, secondary forest, and ravine biotopes and negatively associated with pine forest, houses, and roads. Resting sites had an aggregated spatial pattern from 0 to 900 m, but their distribution was completely random at larger scales. Our study revealed that Central American brocket deer selected places with specific characteristics to rest, at both microhabitat and landscape scales. We therefore suggest that existing habitat be increased by reforesting with native species—particularly Mexican beech forest and oak forest—to improve the deer’s conservation status in the study area.

Introduction

The Central American brocket deer is a vulnerable species, with geographically isolated populations mainly found in mountain cloud forests and evergreen forests (Gallina-Tessaro et al., 2019). The species is currently facing conservation problems due to overexploitation and the loss and fragmentation of its habitats (Muñoz Vazquez & Gallina Tessaro, 2016). In addition to the current scenario, projections of future habitat transformation are not favorable for the species, making it critical to understand its habitat characteristics, particularly of its resting sites, which play a crucial role in maintaining the species (Cuarón, 2000; Acosta, 2011; Findlay et al., 2015).

Resting sites have important effects on survival and fitness; they provide adults and offspring with protection from predators, thermal cover, comfort, and access to food and water resources (Lutermann, Verburgt & Rendigs, 2010; Li, Li & Liu, 2017).

Resting sites allow an animal to hide from predators during inactive periods and breeding and get access to escape routes, in order to reduce predation risk. This is particularly important for prey species such as Central American brocket deer that is the most important prey for cougar (Puma concolor), an essential part of the diet of jaguar (Panthera onca) and an occasional prey of tayras (Eira Barbara; Foster et al., 2010; Mello et al., 2021). Domestic dogs (Canis lupus familiaris) also attack and kill Central American brocket deer (Muñoz Vazquez, 2013).

Resting sites also provide thermal cover, defined as coniferous or deciduous overstories and or plant understories that protect an animal from microclimatic extremes of heat and radiation mainly during solar noon. This protection decreases the metabolic costs related to heat dissipation, as well as providing shelter from rain and blocking wind, buffering against variation in temperature and moisture (Demarchi & Bunnell, 1993).

As well as security, resting sites must offer comfort to an animal to assure its well-being (Erdtmann & Keuling, 2020). Comfort behaviors include the selection of the resting surface. For instance, some deer species that inhabit clay-eroded ravine landscapes select a flat spot so that they do not slide off the resting site, which they scrape by pawing the soil and remove branches so the resting site itself becomes more concave (Mysterud & Ostbye, 1995).

Access to food and water sources are also potentially important aspects of resting site habitat. It has been observed that most territories contain patches of high-quality food that limits deer distribution (Carranza, FernandezLlario & Gomendio, 1996). The diet of the Central American brocket deer varies among regions and ecosystems; in the tropics it behaves like a specialist frugivore, while in the mountain cloud forests it behaves like a browser with a broader consumption spectrum (Weber, 2005; Villarreal-Espino-Barros et al., 2008; Flores-Vazquez, 2021). Access to water resources, on the other hand, is important even in places where water is not limited (e.g., in mountain cloud forests, which retain moisture nearly year-round due to horizontal rain). For instance, some large herbivores that inhabit this ecosystem locate their resting sites near to some ponds that contained water all year for cooling off during the hottest hours of the day (De la Torre et al., 2018).

Research on the Central American brocket deer has primarily considered their spatial distribution and basic ecology, leaving the habitat requirements for resting sites virtually unknown (Weber, 2005; Muñoz Vazquez & Gallina Tessaro, 2016). Here, we describe the characteristics of Central American brocket deer resting sites. We did this at two different scales: microhabitat and landscape (Johnson, 1980). We hypothesized that the distribution and structure of resting sites would be driven by variables related to thermal cover, protection from predators, comfort, access to food and water resources and habitat preferences. We predicted that Central American brocket deer resting sites would follow an aggregated pattern and that they would have some specific characteristics. At the microhabitat scale, we expected greater thermal and hiding cover, flatter surfaces and more comfort behavior signs, more edible plant species, and shorter distance to water in resting sites compared to randomly selected control sites. At the landscape scale, we predicted (based on Central American brocket deer and other deer species habitat preferences) that resting sites would be most frequent in beech forests, at elevations above 1500 m, with flat or almost flat slopes and with hillsides oriented to the north.

Material and Methods

Study area

The study was carried out in the mountain cloud forest of San Bartolo Tutotepec, which spans over 6070.1 ha in eastern Hidalgo, Mexico (UTM 572661 2261171, 582857 2261149, 572639 2255120, 582880 2255098; Fig. 1). The forest forms part of the priority region “Mountain cloud forests of the Sierra Madre Oriental” and the Ecoregion “Montane Forests of Veracruz” (CONABIO, 2010). The climate is temperate and humid with two seasonal periods throughout the year: a dry season from October to May and a rainy season from June to September (Peters, 1997). The annual rainfall is 1,200 to 2,000 mm, and the average temperature is approximately 12 °C to 18 °C. It ranges in elevation from 200 to 1,944 m and has a rugged topography, with steep slopes where rivers and streams run between the pronounced ravines, emptying into the river Chiflón (CONAGUA, 2012). The hilly countryside is covered by tropical mountain cloud forest fragmented by agricultural lands and pastures. It contains relict-endemic and endangered tree species such as Magnolia schiedeana, Fagus grandifolia subsp. mexicana, Quercus delgadoana, Q. trinitatis, Q. meavei, Symplocos coccinea, Styrax glabrescens, Turpinia insignis, Persea spp. Several tree fern species, including Cyathea fulva, Dicksonia sellowiana var. arachneosa and Alsophila firma, inhabit steep slopes. The understory is mainly composed of Miconia glaberrima. Central American brocket deer share habitat with other mammals such as coyote (Canis latrans), gray fox (Urocyon cinereoargenteus), jaguarundi (Puma yagouaroundi), ocelot (Leopardus pardalis), margay (Leopardus wiedii), American hog-nosed skunk (Conepatus leuconotus), hooded skunk (Mephitis macroura), long-tailed weasel (Mustela frenata), ringtails (Bassariscus astutus), white-nosed coati (Nasua narica), kinkajou (Potos flavus), raccoon (Procyon lotor), nine-banded armadillo (Dasypus novemcinctus), Eastern cottontails (Sylvilagus floridanus), lowland paca (Cuniculus paca) and Mexican red-bellied squirrel (Sciurus aureogaster) among others (Huerta-Valdez, 2017). It is important to mention that until recently it was thought that its natural predator, the cougar (Puma concolor), had been extirpated from the area, but recent observations confirm its presence (A Cruz-Oropeza, 2021, unpublished data).

Figure 1 Geographical location of the San Bartolo Tropical Montane Cloud Forest of the Sierra Madre Oriental in eastern Mexico, where we analyzed the distribution and characteristics of Central American brocket deer resting sites from 2017 to 2019.

The study area has been occupied by humans since the 10th century, when the Toltec civilization inhabited the area. Currently there are 12 Otomí communities in the region, ranging from 40 to 258 inhabitants, whose main economic activities are agriculture and extensive livestock production (Muñoz Vazquez, 2013).

Resting site attributes

We conducted eight bimonthly field surveys between November 2017 and March 2019 with four transects per field season (32 in total). Each transect was 500 m long and were distributed uniformly within the part of the study area that is suitable for brocket deer, according to Muñoz Vazquez & Gallina Tessaro (2016); i.e., avoiding human settlements, pastures and crops. The starting points of the transects were randomly selected a priori and the direction was always north-south. The minimum distance between transects was 200 m in order to cover as much habitat as possible. Each every transect was visited once by four people who directly searched for resting sites (Fig. 2).

Figure 2 Location of sampling transects and identified resting sites of Central American brocket deer in San Bartolo, Hidalgo, México 2017–2019.

Transects are illustrated as dot lines while resting sites are represented by white dots.

We identified resting sites based on the presence of fecal pellets following Aranda (2012), which sometimes formed latrines. We also searched for footprints, scrapes (soil disturbed by deer pawing at the ground), and browsed plants (Fig. 3). Whenever we observed a resting site, we recorded its coordinates using a GPS.

Figure 3 Illustration of the method used to characterize microhabitat of Central American brocket deer in San Bartolo, Hidalgo, México 2017–2019.

(A) Identification of the resting site, slope from the ground, distance to the nearest water resource and 10 m transect to each cardinal point. (B) Illustration of the method to measure concealment cover for fawns and adults. (C) Illustration of the method to measure canopy closure.

To describe the resting site, we assessed habitat characteristics at two different spatial scales: the microhabitat (specific surface that was used for resting and the 20 m2 surrounding the resting site) and the landscape (the 30 m2 surrounding the resting site).

Microhabitat

Resting sites were characterized by recording vertical and horizontal thermal cover, protection from predators for fawns and adults, comfort signs, and availability of food and water. First, we performed 20 readings of canopy closure (five in each of the four cardinal directions) between 9:00 am and 12:00 pm using a densiometer to estimate vertical thermal cover (Model C, Lemmon, 1957, Fig. 3C). Second, we followed the Canfield method along four 10 m-long transects (beginning at the resting site and progressing in each cardinal direction), to record all understory vegetation cover ≤2 m in height. These surveys were used to calculate the understory density and coverage height as measures of horizontal thermal cover (see Canfield, 1941).

We measured concealment cover following Griffith & Youtie (1988) at 10 m from the resting site; the only modification was that we considered the cover closest to the ground (“first segment”, 0–50 cm) as protection from predators for fawns and the cover in the second (50–100 cm) and third segments (100–150 cm) as protection from predators for adults (Fig. 3B). We also counted deer footpaths using a manual counter (Base Mount Tally Counter), which we considered the potential escape routes from predators.

As comfort signs, we measured the slope from the ground by using a clinometer (SUUNTO), and we registered the presence/absence of scrapes and the tree and/or tree fern species under which the site was located to determine whether there was a preference for a particular species of tree/tree fern for resting sites.

To evaluate food availability, we identified the plant species along the transect and compared to the to the lists of plant species consumed by deer compiled by Villarreal-Espino-Barros et al. (2008) and Flores-Vazquez (2021) and calculated the cumulative importance value of the edible plant species with the following formulas:

v.i.acum=v.i.sp1+v.i.sp2+…+v.i.spn

v.i.ediblespecies=understorydensity+understorycoverage+frequency.

Finally, to record access to water, we measured the linear distance from the resting site to the nearest water source (e.g., permanent and ephemeral ponds, creeks), using a measuring tape (Fig. 3).

Landscape

Since we hypothesized that the habitat preferences of the Central American brocket deer influenced the selection of its resting sites, we used four landscape-scale variables to describe resting sites. Type of biotope was derived from a Landsat 8 OLI image, using an Iso cluster function with unsupervised classification to obtain a biotope type layer that was categorized into the following discrete classes: beech forest, oak forest, secondary vegetation, pine forest, rainforest, ravines, houses and roads, and grazing areas (areas devoid of native vegetation and dedicated to livestock grazing). Elevation was derived directly from a digital elevation model (DEM), and aspect and slope that were calculated with the Surface toolbox in Arc Map 10.3 from the DEM. Erdas Imagine software v.14.0 was used to reproject the Landsat and DEM images and to perform atmospheric and radiometric corrections (Intergraph Corporation). Pixel size for both images was 30 m2.

For each resting site surveyed, we established a paired control plot where we measured the same microhabitat and landscape attributes. Control plots were selected by measuring at the point 50 m from the identified resting site in a randomly selected cardinal direction.

Statistical analyses

We first tested whether the explanatory variables were normally distributed. Spearman rank tests were used to evaluate correlations (rs ≥ —0.7—). Then, multiple logistic regressions were used to differentiate combinations of variables associated with the resting sites. We chose this analysis because multiple logistic regressions are especially useful when the data consist of both discrete and continuous variables (Powell, 2000).

Resting site distribution

We used a point pattern analysis to describe resting site distribution. First, we used the resting site locations to construct a planar point pattern (ppp), then used the Kernel-smoothed intensity to measure the mean number of occurrences per unit at a point (u) defined by λ (u). Finally, we performed first-order characteristics analysis of a spatial process for a general location (s; Eq. (1)):

(1) λbs=1Cbs∑i=1nKibs−xi

where Kb() is a Kernel with band b > 0, and Cb() is an edge correction factor (Yang et al., 2007).

We performed a quadrat count test to determine whether there was complete spatial randomness. Once we determined that the pattern was not random, we calculated the inhomogeneous K and L functions (Eq. (2)): (2) K ˆinhomr=1W∑i ∑j≠i1xi−xj≤rλ ˆxiλ ˆxjexi,xj;r

where e (u, v, r) is an edge correction weight and λ ˆu is an estimate of the intensity function λ(u).

Results

We found and characterized 43 resting sites and their corresponding control plots. It is important to note that this is the first time that latrine formation is reported for Central American brocket deer. At the microhabitat scale, we observed that the probability of being a resting site in the area was positively associated with vertical thermal cover, concealment cover, the number of escape routes and the presence of edible plant species and negatively associated with the distance to water resources. Resting sites were found on flat and nearly flat surfaces, and scrapes were only recorded at resting sites (Table 1).

Table 1 Microhabitat and landscape parameters that were selected to describe resting sites and control plots for the probability of resting sites versus control plots for Central American brocket deer.

Parameter	Average	Estimate	SE	Odds ratio	
Microhabitat	Resting sites	Control plot				
Canopy closure (%)	98.19	68.14	0.57	8.34	1.76	
Understory density (ind/m)	0.50	0.52	−2.95	273.84	0.05	
Understory height (m)	1.00	0.39	−0.53	179.53	0.59	
Concealment for fawns (%)	99.42	80.70	−0.02	2.77	0.98	
Concealment for adults (%)	71.31	65.16	0.01	2.64	1.01	
Escape routes (n)	4.14	2.53	0.47	44.71	1.6	
Slope from the ground (∇)	170.58	36.49	0.31	2.29	1.37	
Scrapes	0.37	0.00	1.19	121.82	3.31	
Tree/tree fern DAP (cm)	91.03	30.70	−0.005	0.52	0.99	
Edible plant species (v.i.a)	0.89	0.71	1.19	79.34	3.29	
Distance to water (m)	32.78	131.35	−0.11	0.03	0.89	
Landscape						
Type of biotope						
Beech			2.55	183.9	12.86	
Oak			3.64	177.14	38.47	
Secondary			2.84	224.7	17.2	
Pine			−2.36	230.17	0.09	
Rainforest			0.06	253.24	1.06	
Ravines			1.53	322.73	4.6	
Houses and roads			−1.17	475.77	0.31	
Grazing			−6.52	321.22	0.001	
Elevation (m.a.s.l)	1846.30	1884.4	−0.003	0.56	0.99	
Aspect	180.39	186.36	−0.0001	0.41	0.99	
Slope (∇)	14.34	26.83	−0.006	3.37	0.99	

At the landscape scale, we observed that the probability of being a resting site was positively associated with beech forest, oak forest, secondary forest and ravine biotopes and negatively associated with pine forest, houses and roads and grazing area biotopes (Table 1).

The highest density of resting sites occurred in two “hotspots” located at the center of the study area (Fig. 4). K and L function graphs showed that the empirical curve was higher than the theoretical curve at distances up to 900 m (Fig. 5). Therefore, we determined that the resting sites had an aggregated spatial pattern from 0 to 900 m, while their distribution was completely random at larger scales.

Figure 4 Representation of the spatial point pattern analysis of Central American brocket deer resting site distribution in San Bartolo, Hidalgo, México 2017-2019.

Kernel-smoothed intensity goes from purple to yellow where the function detected an aggregation of resting sites, which are represented with red crosses.

Figure 5 Generalized L. function for the spatial pattern analysis of Central American brocket deer resting site distribution in San Bartolo, Hidalgo, México 2017–2019.

The shaded area shows envelopes from 99 simulations of each model, while the solid black line represents the empirical function from the fitted model and the dotted line shows the mean of the function from the fitted model. When the empirical function (solid line) goes above the mean function (red dotted line) means there is an aggregation of resting sites greater than expected by chance.

Discussion

Our results support the idea that resting sites are important places for Central American brocket deer, since they provide adults and offspring with thermal cover, protection from predators, comfort, and access to food and water resources. Central American brocket deer selected resting site locations with high canopy closure, probably seeking protection from extreme heat and radiation during solar noon and from the rain and excess moisture in the extremely rainy environment of the mountain cloud forest (Demarchi & Bunnell, 1993).

Our results also showed the importance of concealment cover and the presence of escape routes, suggesting that landscapes that offer more protection contribute to the avoidance of predators. This trend has also been reported for other deer species, such as fawn and adults of white-tailed deer (Huegel, Dahlgren & Gladfelter, 1986; Gallina et al., 2010). In our study area, we detected the presence of the main natural predator of Central American brocket deer, Puma concolor, and of free-ranging and feral dogs that can also act as predators (Foster et al., 2010; Huerta-Valdez, 2017; Christen, Janko & Rehnus, 2018).

Not surprisingly, deer chose resting sites located closer to streams and rills than expected at random. Central American brocket deer have an affinity for water bodies. The species was originally named temamazame, which means “deer that likes water,” by Hernandez in his expedition to Nueva España during 1571–1574 (Hernandez, 1651). Also, deer were recorded swimming across the Lacantun river during the dry season when the waters were shallow (Naranjo & Bodmer, 2007). It is very likely that deer use the water bodies for cooling off during the hottest hours of the day and to prevent ectoparasites (De la Torre et al., 2018; Delgado-Martínez et al., 2018; Lira et al., 2018). This behavior coincides with that observed in brocket deer in the Chiquitano forest of Bolivia, where the species preferred riverine forests where water was available year-round to other habitats where ponds and running streams are present only during the rainy season (Rivero, Rumiz & Taber, 2005).

Central American brocket deer also prefer flat spots (≈180°) to rest, and the presence of scrapes by pawing the soil were common. This was despite the fact that resting sites were located in areas where the overall terrain was steep; this suggests that they chose locally flat sites to avoid sliding off of the resting site, which could be interpreted as a comfort sign (Erdtmann & Keuling, 2020). It has been observed in other deer species that beds occupied over the longer term are more frequently scraped than beds used for a short time, and scraping behavior may also function in olfactory communication (Mysterud & Ostbye, 1995; Black-Decima & Santana, 2011; Hearst et al., 2021).

Our results showed that biotope was the most important variable in resting site distribution. Similarly, Garcia-Marmolejo et al. (2013) found that landscape composition was the single most important variable contributing to potential distribution of M. temama in the Huasteca region of San Luis Potosí, Mexico. Despite the availability of different biotopes, a relatively high percentage of resting sites were found in Mexican beech forest habitats. Remnants of Mexican beech forest are considered hotspots with high connectivity and low disturbance in our study area (Rodríguez-Ramírez, Sánchez-González & Ángeles Pérez, 2016). Additionally, secondary vegetation clearly played an important role in resting sites, coinciding with other studies where deer are distributed in secondary vegetation near well-preserved forests (Bello-Gutiérrez, Guzmán-Aguirre & Chablé-Montero, 2004). Ravines were also important sites for Central American brocket deer resting sites, in our study area ravines, locally known as “jewels” are places where the mountains come together and create high humidity microhabitats that favor the presence of dense vegetation, as well as steep and sloping terrain, which generates good hiding places for the deer and are sometimes used as safe trails from potential predators such as cougar (Puma cursivas) or dogs. Other mammals, such as the common genet (Genetta genetta) in the Mediterranean, have been shown to use ravines similarly as safe passages and resting site locations (Camps, 2011).

Regarding the distribution of resting sites, we found that Central American brocket deer were selective. Resting sites were only found in the center of the study area, in accordance with results across southeastern Mexico that show that the species restricts its distribution to areas far from the forest edges, with only occasional excursions to edge habitats. Thus, the current study supports previous findings showing that Central American brocket deer may be considered a habitat specialist (Bello-Gutiérrez, Guzmán-Aguirre & Chablé-Montero, 2004; Weber, 2005; Weber, 2008).

Conclusions

Our study revealed that Central American brocket deer select places with specific characteristics to rest, at both the microhabitat and landscape scales. The most important parameters at the microhabitat scale were food and water availability, vertical thermal cover, concealment cover, and slope from the ground, while at the landscape scale the most important variables were the type of biotope (positively associated with beech forest, oak forest, secondary vegetation and ravines, and negatively associated with houses and roads and grazing areas). In our study area, there are only a few forest patches that meet Central American brocket deer resting site requirements; most of the habitat is unsuitable for this behavior due to disturbance, including intense agricultural and livestock activity. This scenario has been observed in most of the studies of this species, and Central American brocket deer habitat is frequently highly fragmented. Therefore, we recommend increasing the existing habitat by reforesting with native species, especially in Mexican beech forest and oak forest, to improve the species’ conservation status in the area.

Supplemental Information

Supplemental Information 1 Data obtained in the field on the characteristics of the beds

Characteristics of the resting sites (e.g., canopy coverage, slope orientation, herbaceous cover).

Click here for additional data file.

We are grateful to Emiliano Donadio, Patricia Black Decima and one anonymous reviewer for improving the manuscript. This paper constitutes part of the doctoral research of BMV. BMV gratefully acknowledges E. Chanes for his support in every aspect of this project.

Additional Information and Declarations

Competing Interests

Author Contributions

Data Avaiability

The authors declare that there are no competing interests.

Brenda Muñoz Vazquez conceived and designed the experiments, performed the experiments, analyzed the data, prepared figures and/or tables, authored or reviewed drafts of the paper, and approved the final draft.

Sonia Gallina Tessaro and Livia León-Paniagua conceived and designed the experiments, performed the experiments, analyzed the data, authored or reviewed drafts of the paper, and approved the final draft.

The following information was supplied regarding data availability:

The raw data is available in the Supplemental File.

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
