# Peer review of "Characteristics of Central American brocket deer resting sites in a tropical mountain cloud forest in eastern Mexico"

_PeerJ, doi:10.7717/peerj.12587_

## Round 0.1 · original submission · Major Revisions

Please revise your manuscript to address the reviewers' concerns. Note that quite substantial revisions are necessary before your paper can be considered for publication in PeerJ.

·

Basic reporting

no comment

Experimental design

no comment

Validity of the findings

no comment

Additional comments

Ref: Manuscript # 61216-v0

In this manuscript Vazquez et al. evaluate the attributes at the landscape and microhabitat levels that drive the selection of resting sites by Mazama temama. This work is based on commendable sampling and analytical efforts and provides important information on a barely studied species.

Having said that, I think this manuscript needs to be improved before publication. My main concerns relate to the way the hypothesis has been laid out and even most importantly the context within which the data is discussed. Finally, I, of course, understand the challenges of writing in a second language, but the manuscript would benefit from a thorough review of the writing (some comments and suggestions about this issue in the attached pdf).

Main comments

Introduction

Lines 60-61 I suggest improving the hypothesis. I think that it could be something like “we hypothesized that the distribution and structure of resting sites would be driven by variables related to access to food, water and protection from predators” From here authors could derived specific predictions about the variables affecting the distribution and structure of resting sites (e.g., distance to the water in resting sites would be shorter than for random sites). I encourage the authors to polish their working hypothesis and present predictions; by doing this they would certainly improve the quality of the manuscript. Moreover, such improvement will help them to structure the manuscript in clearer and more concise way.

In lines 149-150 authors state that they used several variables to describe thermal cover; the problem here is that in the introduction authors only mention food, water, and protection from predators; so, thermal cover must be included in the intro.

Methods

There is a missing piece in the methods. Basically, there is no mention to how authors designed the transect survey. Starting points and direction were randomly selected? Please include this information.

Methods to evaluate microhabitat need to be better organized. Begin by clarifying which variables were used to describe water, food, and protection. For instance, “we evaluated access to the water by measuring the following variables…”, and so on for each resource. Also, I could not find any variable linked to food; please explain why + variables related to food are missing.
Authors state that they calculated the understory density, coverage, height, and species richness; and that these variables were then used to describe thermal cover. How is species richness related to thermal cover? Also, there is no mention to thermal cover in the introduction.

I think authors might want to explain in a more friendly way what was specifically tested with their methods; in some cases, this is done nicely, but in other cases I found statements like “we calculated the homogeneous K and L…” difficult to understand because I do not how they contribute to the story.

Finally, it is not clear why you looked at scraps, branches removed, compacted litter, and preference for bedding below a tree and/or tree fern species. How does this part of your analysis fit the introduction?

Results

Just difficult to follow and could be more informative. As a reader what I would like to see in the results are effect sizes and odds. For instance, cover was XX times higher in resting sites compared to random sites. Or the odds of finding a resting site increased by XX with every meter closest to a water source. Those are, I think, the kind of data that highlight the importance of your work. In fact, that sort of data are the ones that could have an impact on conservation and management actions.

Discussion
In the discussion, please delete the first paragraph (lines 202-208). There is no need to spend a whole paragraph trying to defend your sample size.

Authors mention core areas several times; but core areas need to be defined in some way. What represents a core area for your species, and why?

Authors refer to a shift (line 220), but they are referring to what it seems some level of habitat selection (“Despite the availability of different vegetation types, a relatively high percentage of resting sites were found in Mexican beech forest habitats”).
Is this the shift that authors mention? If so, that´s not a shift.

On the other hand, if deer shifted habitat use (¿?), then, they are not habitat specialists as you claim. Again, in lines 228-240, authors state that evergreen forest was avoided but because of human activity and the fact that such habitat has been heavily impacted. So, deer is not selecting against evergreen forest because some intrinsic variable; it is avoiding it because it is dangerous. Or even worst, you did not detect resting sites there because animals are just being poached. So, it seems that the context does not support your claim about habitat specialization of this deer. All this confusion must be clarified before publication because is a clear weakness in the interpretation of the results.

Lines 225-227 how is that the use of ravines reflect antipredator behavior?

Paragraph beginning in Line 257 mentions the importance of thermal and concealment cover, but only discusses concealment and ends mentioning that “As a forest specialist, it is not surprising that this species prefers forest habitat to rest” So, how is this final statement related to thermal and concealment cover?

Lines 265-270 introduce something defined as “general habitat” Not sure why this is important. I suggest authors delete this paragraph. Similarly, lines 271-275 mention some behavioral attributes of the species that make deer difficult to observe and highlight the importance of tracks to detect resting sites and how thorough the survey was. How is this related to your discussion of the results?

Minor comments

Line 27-28 Habitat features might define resting sites, but they are not resting sites. Please rephrase this sentence.
Line 32 “Resting sites were aggregated at distances below one km.” Unclear, please rephrase.
Lines 32-33 “Vegetation type and elevation affected resting site distribution at a landscape scale”. Authors need to state how these variables affected the distribution of resting sites.
Line 33 “all resting sites displayed similar values” What are the values?
Lines 38-40 “preserving Mexican beech forest and oak forest and protecting deer from poaching in core resting site areas are particularly crucial” Why? Can you link resting sites to some key demographic parameter?

In general, the abstract needs to be more informative.

Lines 54-55 delete the sentence “Several authors have described the characteristics of resting sites for many species, but this does not include Central American brocket deer” Start the paragraph with the second sentence but remove “this species” and include Central America brocket deer instead.

Line 79 what is the difference between bedding and resting sites? If none, then use only one term. If there is a difference, then define each type of site.

Lines 111-112 How is that roads and villages are classified as types of vegetation? I suggest changing the name of the variable from Vegetation type to something else.

Lines 114-116 Authors converted a continue quantitative variable into a categorical one; by doing so, they lose a lot of information. Please support your decision.

Lines 127 and 129 Please explain what the role of cartesian coordinates is in these 2 models.

Line 139 please explain how deer footpaths and slope are connected to water, food, or protection requirements.

Lines 287-288 “...alleviate potential human-wildlife conflicts” This manuscript is not about human-wildlife conflict. Please delete.

Legends for all tables and figures need to be more self-explanatory. Moreover, several components of the tables are difficult to interpret. For instance, Table 1 what are Pars and M3value?; Table T2, need to define what each C is, and also define the interaction term.
Figure 6, axes labels are illegible; in some figures I cannot figure it out what the grey shaded areas are; in general, what are you showing with those figures? Explain in the legends.

Figure 3, I am not sure why it is important; I would suggest deleting it. Green scale almost impossible to see. Also, it seems that you are using different font sizes or styles here (e.g., vegetation vs hillside and elevation). Is elevation in meters (see the color key)?

·

Basic reporting

The English is generally good but could be improved with revision by a native speaker.
Some references appear to be incomplete; they may be doctoral theses, but it is not clear from the information provided. The reference Aranda(2012) (line 84) is missing from the bibliography.
The article structure, figures and tables correspond to the way an article should be written. Raw data is supplied.
It is self-contained wih relevant results to hypotheses.

Experimental design

It is original research within the Aims and Scope of the journal. This manuscript deals with a study of resting places in the Central American brocket deer (Mazama temama) in Mexico. Given our lack of knowledge of the biology of all species of brocket deer, it is a welcome addition to our understanding of the species.
Methods: Resting sites: more detail should be provided about the methodology used in the field. The authors state that they sampled along 32 transects of 500 m each over a period of almost 1 ½ years. They do not say how the transects were laid out, what separation was between them, whether each transect was sampled once or more often, how often observations were made or in what seasons. They do not indicate the seasonality of the study area, whether it has a wet and dry season or relatively warmer and cooler seasons.

Resting site distribution: the authors state that they compared the number of pixels in each class of vegetation with the number of pixels containing resting sites. Could they clarify the size of pixels they are talking about so that we can understand the scale of the habitat that they are comparing?
Model fitting and diagnostics: the authors state that they used Poisson models to explain the distribution. Are they talking about Poisson regressions for GLMs? This is not clear, and as these are very complicated statistics for the general reader, they should clarify exactly what type of model they are using. The description of the difference among the four models themselves is clear. However, Lurking variables and second order K functions are not familiar concepts and a little explanation would be in order.
Microhabitat: The descriptions and variables measured are well done, but more descriptive detail would be very helpful. In line 151 they say they looked at “scraps”; I assume they mean scrapes. What kind of scrapes are these? Are they marks on trees, described in gray and red brockets, where the deer scrape the bark with their incisors and then rub their foreheads or are they scrapes on the ground as made by pawing the ground? They mentioned the presence of fecal pellets as part of the identification of bedding sites (lines 79-80), but they do not mention the presence of latrines near the bedding sites. The use of latrines, probably for scent-marking has been described in red and gray brocket deer (M. americana and M. gouazoubira) (Black-Decima & Santana, 2011 (Acta Theriol (2011) 56:179–187 DOI 10.1007/s13364-010-0017-6); Rivero et et al 2004, 2005 (Eur J Wildl Res (2004) 50: 161–167
DOI 10.1007/s10344-004-0064-x; Mammalia 69 (2) : 169-183 ). This information would be helpful for a general understanding of the biology of brocket deer.

Validity of the findings

Results: Resting site distribution and Landscape variables. The distribution of resting sites is very interesting (Fig,2A). However, one immediately wants to see a map of the study area and compare it with this distribution. It takes awhile to realize that these maps seem to be in Fig. 3, as the coordinates coincide. This information should be in the figure legend for Fig. 3 and should be referenced in the text near the description of the resting sites. The legend in Fig. 3A could be improved as the lines and colors are very close together and hard to distinguish. A map of the study area with the resting sides superimposed would be helpful.
Model fitting and diagnostics: The model fitting seems to be fine, especially if there is a sentence about Lurking variables in the methods section or here.
Microhabitat: The analysis is good. Again, it would be very interesting to know whether latrines or marked trees were found near the resting sites. The statistics also corroborate their findings. Again, inhomogenous second order K functions are not common knowledge and a little explanation is in order.
The statistical comparion is clear, and Fig.6 is potentially clear and valuable. However, the quality of the figure is poor; it is too small, it is impossible to read and it is too low resolution.
The conclusions are valid and well linked to the original question.
Discussion and possible speculation:
With respect to th discussion, the points raised are clear and valuable. However, it would be interesting to know if the authors observed any instances of scent-marking of trees and latrines to add to the general biology. It would also be interesting to know if they made any estimates of the number of deer that used these resting sites, considering previous abundance measurements and considering the solitary nature of these deer. Reyna-Hurtado & Sanchez-Pinzon (2019) consider the density of M. temama in the Chakamul region of Mexio to be 0.9-1.5 deer/km2. This would correspond to around 1 individual in the smaller region of Fig. 2A and 3-4 in the larger region. These may represent the home ranges of very few individuals. Further studies using camara traps might yield valuable information about this little-known species.

Additional comments

I am happy to see field work on brocket deer and I hope the author continues her work in more detail.

Reviewer 3 ·

Basic reporting

Introduction
There is way too little information in the introduction section. The authors need to establish why this study is important and how it has ecological implications for this species. A much more thorough review of the literatures is necessary for publication. There should be at least 4 paragraphs detailing the importance of this topic based on extensive literature review. Paragraphs need to be a minimum of 3 sentences.
Methods
The study area description is also severely lacking. The writing is poorly structured and the information about the area is at a bare minimum. What about predator presence? This paper is about resting sites, which are important for survival, yet there is nothing about the potential predators in the area. I recommend including more information about the area regarding seasonality, precipitation, temperatures, topography, population characteristics, predator presence, etc. In its current form, I know very little information about the area.
The writing needs major work and is not suitable for publication, throughout the manuscript. See manuscripts in this journal and others for examples.
The results need to be much more detailed and really describe all the major findings of each analysis.
The authors state that the sites were located in a year but say otherwise in the methods section.
The discussion is much more thorough than any other portion of the paper. However, the authors need to tie everything together at the end better. Describe why this is pertinent information for the management of the species.
Tables need to be reformatted so everything is very clear. Also, where are the headings for tables?
The figures need to be redone so that each axis is clear and wording is not clumped along axes. Figure 2 needs a scale. Where are the headings/descriptions?

Experimental design

The limited information regarding the collection of data seems appropriate, although more information is needed to truly assess this. How were the transects distributed? Were they covering an equal representation of habitat types available in the study area? What time of year were they conducted?
Why did you use a point pattern analysis? This needs to be justified. Many would just use logistic regression frameworks to compare the used vs random locations of resting sites. See Smith et al. 2015 in The Journal of Wildlife Management for an example of more common analyses used for this type of work. The descriptions of the analysis are bare minimum and don’t really allow the reader to understand exactly what you are doing. The authors need to be very thorough with describing the analysis and the math behind it.
What is the purpose of using NDVI? If this is a forested area, NDVI is just going to show the values from the forest and not the understory. This is why it is important to thoroughly describe your study area. NDVI in treed areas is not useful for studying ungulates unless the trees are a primary source of vegetation. If they are not, then NDVI values need to be regressed against a tree cover layer so that they are adjusted. However, in dense canopy forests, this is still not going to give an accurate representation of vegetation quality in the area. Given the low number of sites, vegetation sampling at each site would have been much more valuable. As it stands, NDVI appears basically useless for this analysis.
There are way too many vegetation types for the amount of sites you have. When dealing with small sample sizes, categorical variables severely limit what you can accomplish. Also, roads and disturbances should be treated as distance to variables. Similarly, treat aspect as a continuous variable. Turn aspect into cosine and sine of aspect variables using the raster calculator. You can also turn aspect into a single, continuous variable. Elevation should not be set as a categorical variable. Make it continuous. Keep in mind this is will help strengthen your models.
How large are pixel sizes for each of your layers and how large are your resting and random sites?
How were your microhabitat variables used to describe thermal cover?
What does “looked at scraps, branches removed, etc” mean in regards to this analysis?
How are you analyzing the vegetation portion of the analysis?
Were variables tested for correlation? It’s also not appropriate to just test for all variables in a single model. The authors need to test many different combinations because variables will interact differently with each other and could lead to skewed results.

Validity of the findings

The results are poorly stated and are hurt in part by the lack of explanations within the methods sections.
The discussion does a good job of relating the results to literature and the ecology of the species, but it is difficult to determine the validity of these findings when the methods are so poorly constructed.

Additional comments

Overall, I think there is some interesting and worthwhile data present in this study, however, in its current format it is not publishable. Significant adjustments need to be done to the manuscript’s writing style and much more clarity needs to be provided throughout. I thought the discussion was the most structured and well written portion of the paper, while the introduction was severely lacking background information. The methods are very difficult to process in their current forms. It is not clear at all what you are trying to accomplish and there are many flaws in the habitat analyses. The analyses as stated do not seem appropriate for this work and need to be revised and at a minimum made clearer. With the right analyses, this data is certainly publishable. Consider using logistic regression analyses for the habitat portions.

---

## Round 0.2 · Minor Revisions

The reviewers have identified a number of issues that remain to be fixed. Please revise your manuscript again to address their concerns.

·

Basic reporting

Ref: Manuscript # 61216-v0-Rev

I appreciate the efforts made by the authors to improve the quality of the ms following my advice and the advice provided by other reviewers.

Some comments about the reviewed version,

Lines 42-43. What is the evidence that shows that current conservation problems result in part from consumption during the Pleistocene? If there is no evidence, please remove this statement.

Line 48 – “Resting sites have important effects on survival and fitness; they provide adults” provide references that support this statement.

Lines 59-60 delete the word “apex”; just say that domestic dogs also kill deer.

Line 66 – Why resting sites must offer food?

Lines 126-132 So, you surveyed a total of 32 transects over 8 field seasons. Basically 4 transects per field season. Is this correct? If yes, write it in the ms, if not also explain (maybe you could state on average 4 transects per field season or numbers of transects surveyed ranged between X and X per field season)

Line 202-203 You repeat 3 times (results, discussion line 222, and conclusion) that you found for the very first time a latrine for the species. This is completely anecdotical and unimportant. I suggest you mention this finding one time maybe in the results, and that should do it.

Line 209-211 you state that the “probability to find a resting site was (…) negatively associated with pine forest, houses and roads and grazing” So a pixel occupied by a house showed a low probability to present a resting site. But pixels were 30 m2; so 6x5 m basically the size of a very small house. I am sure you were not expecting to find a resting site in a house. Why are houses part of the analyses? The same for roads. Also, you need to define “grazing”. Did you mean grazing areas?

Line 222-226 Please delete; your work is not about latrines and their function.

Line 236 delete “apex”. In fact, if pumas inhabit the area, dogs are not the apex predators.

Line 238-242 delete, your work is not about diet.

Lines 243-250 How do you interpret this result?

Line 272 – dogs were introduced previously; delete scientific name.

Line 283 – Unclear. Please cite work where managers use “habitat limitations” to protect populations.

Lines 294-295 – Authors conclude “Therefore, we suggest preserving the existing forests, with a special interest in Mexican beech forest and oak forest to improve the
species conservation status.” How is that preserving what it exists will improve the conservation status of the deer? Please explain.

Please check the English; it has been improved, but there sentences with verbs in past and present tense, repeated words within clauses, and sentences that are too long and should be broken into 2 or 3 sentences.

Experimental design

see 1

Validity of the findings

see 1

Additional comments

NA

·

Basic reporting

The manuscript has been greatly improved from the original version presented. However, the English is still a problem. Throughout the manuscript there are many grammatical errors such as lack of correspondence of verb forms with subjects and sentences that are so long as to be difficult to understand. The manuscript needs to be corrected by a native English speaker. The literature has been well covered, the background and structure are fine. The analysis has been greatly improved as well as the figures. One small correction: on line 112 the authors refer to jaguarundi (Herpailurus yagouaroundi). The scientific name has been changed to Puma yagouaroundi. The other problem I have is with Figure 5. The legend states that it shows the L function of the distribution of resting sites; however, the title of the graph is Fecal pellet groups (L function). Are they talking about resting sites or fecal pellet groups?

Experimental design

The experimental design is clearer now than in the first version. The description of the methods is greatly improved.

Validity of the findings

This section is fine.

Additional comments

I hope the authors will continue their investigations of brocket deer, as so much is still unknown abouth them.

---

## Round 0.3 · accepted · Accept

Thank you for your revisions.